# Pharmacological Activation of TRPC6 Channel Prevents Colitis Progression

**DOI:** 10.3390/ijms25042401

**Published:** 2024-02-18

**Authors:** Kazuhiro Nishiyama, Yuri Kato, Akiyuki Nishimura, Xinya Mi, Ryu Nagata, Yasuo Mori, Yasu-Taka Azuma, Motohiro Nishida

**Affiliations:** 1Graduate School of Pharmaceutical Sciences, Kyushu University, Fukuoka 812-8582, Japan; knishiyama@omu.ac.jp (K.N.); yu-kato@phar.kyushu-u.ac.jp (Y.K.); mixinya@phar.kyushu-u.ac.jp (X.M.); 2Laboratory of Prophylactic Pharmacology, Osaka Metropolitan University Graduate School of Veterinary Science, Osaka 598-8531, Japan; yta-vet@omu.ac.jp; 3National Institute for Physiological Sciences (NIPS), National Institutes of Natural Sciences, Okazaki 444-8787, Japan; aki@nips.ac.jp; 4Exploratory Research Center on Life and Living Systems (ExCELLS), National Institutes of Natural Sciences, Okazaki 444-8787, Japan; 5SOKENDAI (Department of Physiological Sciences, School of Life Science, The Graduate University for Advanced Studies), Okazaki 444-8787, Japan; 6Graduate School of Pharmaceutical Sciences, Osaka University, Osaka 565-0871, Japan; nagata-r@phs.osaka-u.ac.jp; 7Graduate School of Engineering, Kyoto University, Kyoto 615-8530, Japan; mori@sbchem.kyoto-u.ac.jp; 8Department of Physiology, Graduate School of Pharmaceutical Sciences, Kyushu University, 3-1-1 Maidashi, Higashi-ku, Fukuoka 812-8582, Japan

**Keywords:** canonical transient receptor potential 6, colitis, stress resistance, gut microbiota

## Abstract

We recently reported that transient receptor potential canonical (TRPC) 6 channel activity contributes to intracellular Zn^2+^ homeostasis in the heart. Zn^2+^ has also been implicated in the regulation of intestinal redox and microbial homeostasis. This study aims to investigate the role of TRPC6-mediated Zn^2+^ influx in the stress resistance of the intestine. The expression profile of TRPC1-C7 mRNAs in the actively inflamed mucosa from inflammatory bowel disease (IBD) patients was analyzed using the GEO database. Systemic TRPC3 knockout (KO) and TRPC6 KO mice were treated with dextran sulfate sodium (DSS) to induce colitis. The Zn^2+^ concentration and the mRNA expression levels of oxidative/inflammatory markers in colon tissues were quantitatively analyzed, and gut microbiota profiles were compared. TRPC6 mRNA expression level was increased in IBD patients and DSS-treated mouse colon tissues. DSS-treated TRPC6 KO mice, but not TRPC3 KO mice, showed severe weight loss and increased disease activity index compared with DSS-treated WT mice. The mRNA abundances of antioxidant proteins were basically increased in the TRPC6 KO colon, with changes in gut microbiota profiles. Treatment with TRPC6 activator prevented the DSS-induced colitis progression accompanied by increasing Zn^2+^ concentration. We suggest that TRPC6-mediated Zn^2+^ influx activity plays a key role in stress resistance against IBD, providing a new strategy for treating colitis.

## 1. Introduction

Intestinal homeostasis is maintained by a complex interaction between the microbiota and the host immune system [1]. The gut microbiota produces a variety of metabolites, thereby controlling many biological processes, such as the immune response and intestinal redox homeostasis. The rapid production of reactive oxygen species (ROS) is important to potentiate the immune system when invaded by pathogens, while this ROS production must be immediately countered by endogenous antioxidant systems to prevent undesired chronic inflammation [2]. Disturbances in the gut microbiota and redox balance cause intestinal stress vulnerability, leading to the progression of inflammatory bowel disease (IBD), such as Crohn’s disease (CD) and ulcerative colitis (UC), characterized by chronic inflammation and mucosal tissue damage with recurrent remissions and relapses in the digestive tract [3,4]. Oxidative stress contributes to the induction and progression of UC [5]. The colon becomes infiltrated and activated by leukocytes, neutrophils, and macrophages, resulting in the increased generation of pro-oxidant molecules accompanied by UC progression [6]. Cytokine-induced increases in myeloperoxidase levels also result in ROS production [7]. In contrast, antioxidant enzymes such as superoxide dismutase (SOD), catalase, glutathione peroxidase (GPx), glutathione reductase, and low-molecular-weight antioxidant molecules such as reduced glutathione (GSH) are present in the epithelium of the colon [8]. These antioxidant/reductant systems of the colonic mucosa are essential for the maintenance of intracellular reducing status and the protection of cells from oxidative stress and exposure to electrophiles such as drugs and phytochemicals [9,10].

Zn^2+^, an essential trace element for living organisms, plays an important role in maintaining redox homeostasis and intestinal microbial homeostasis [11,12]. The intestine is the main location for Zn^2+^ absorption and excretion [13]. Zn^2+^ deficiency is common in patients with IBD [14]. IBD patients show altered expression of ion channels and transporters [15,16]. Several biologically important Zn^2+^ transporters and Zn^2+^-permeable channels have been identified, but a druggable therapeutic target protein has not yet been found for the treatment of IBD. Transient receptor potential (TRP) channels are multimodal sensor/activator channels that can permeate Zn^2+^ [17,18]. Members of the canonical TRP subfamily (TRPC1-TRPC7) are considered molecular entities of receptor-activated cation channels [19], and TRPC6 channels are reported to permeate metal ions such as Zn^2+^ and Fe^2+^ in addition to Ca^2+^ and Na^+^ [18,20]. Growing evidence has suggested that TRPC6 channels contribute to the progression of pathological cardiovascular and renal remodeling and that the inhibition of TRPC channel activities is a potential target for several cardiovascular diseases [19,21,22]. We recently reported the opposite finding that the pharmacological activation of TRPC6 channels improves myocardial contractility and heart failure through a Zn^2+^ influx-dependent pathway [20]. Although TRPC6 channels are ubiquitously expressed, including in immune cells and colon tissues, and may contribute to tissue remodeling by promoting cellular proliferation and/or differentiation, it is unclear whether TRPC6-mediated Zn^2+^ influx also prevents the progression of IBD. This study aims to investigate the role of TRPC6-mediated Zn^2+^ influx in intestinal stress resistance. Here, we examine the role of TRPC6 channels in colitis progression using dextran sulfate sodium (DSS)-treated TRPC6 KO mice and explore whether activation of TRPC6 channels prevents colitis progression by maintaining Zn^2+^ homeostasis.

## 2. Results

### 2.1. Expression of TRPC1-7 in CD and UC Patients

To determine whether TRPC1-7 mRNA expression levels are altered in the inflamed mucosa of CD and UC patients, we searched open resources. In CD patients, TRPC1 and TRPC4 mRNA expression levels were increased, while TRPC3, TRPC4, and TRPC6 mRNA expression levels were elevated in UC patients. The TRPC7 mRNA expression level was decreased in UC patients (Figure 1A). Next, we analyzed TRPC1-7 mRNA expression levels in the colon tissues of the colitis mouse model (Figure 1B). We observed an increase in the TRPC6 expression level in DSS-induced colitis mice compared to the control mice (Figure 1B). Conversely, the TRPC7 expression level was decreased in DSS-induced colitis mice compared to the control mice (Figure 1B), and the TRPC5 expression level was not detected in either DSS-induced colitis or control mice. These results indicate that the TRPC6 mRNA expression level is commonly increased in the colon tissues of patients and mice with IBD.

### 2.2. TRPC6 Protects Mice from DSS-Induced Colitis

Although the structural and functional properties of TRPC6 and TRPC3 are very similar, TRPC3 is not permeable to Zn^2+^. To determine whether TRPC6-mediated Zn^2+^ affects the development of intestinal inflammation, we conducted experiments using whole-body TRPC3 knockout (KO) and TRPC6 KO mice in the DSS-induced colitis model. TRPC6 KO mice exhibited more severe weight loss and increased DAI compared to WT mice (Figure 2A,B). TRPC3 KO mice displayed similar results to wild-type (WT) mice (Figure 2A,B). The mRNA expression level of interleukin (IL)-6 in the colon was significantly increased in TRPC6 KO mice compared to WT mice. The mRNA expression levels of tumor necrosis factor (TNF)α and IL-1β in the colon were similar in TRPC6 KO and WT mice (Figure 2C). Zn^2+^ levels in the colon were significantly decreased in TRPC6 KO mice compared to WT mice (Figure 2D). The expression of TRPC6 was observed in the intestinal mucosal layer and muscle layer (Figure 2E,F). These results suggest that TRPC6-mediated Zn^2+^ participates in stress resistance against intestinal inflammation.

### 2.3. TRPC6 Regulates Antioxidant Protein Expression

Zn^2+^ is important for maintaining redox homeostasis [11]. Antioxidant proteins are known to positively or negatively regulate intestinal inflammation [23,24]. We analyzed the mRNA expression of antioxidant proteins in the colon. The mRNA expression levels of nuclear factor-erythroid 2-related factor 2 (Nrf2), SOD1, peroxiredoxin 3 (Prdx3), and sulfiredoxin 1 (SRXN1) in the colon isolated from intact TRPC6 KO mice were significantly increased compared to those in the colon isolated from intact WT mice (Figure 3A–D). The mRNA expression levels of antioxidant proteins in the colon isolated from DSS-treated TRPC6 KO mice were equivalent to those in the colon isolated from DSS-treated WT mice. These results suggest that TRPC6 channel activity negatively regulates the reducing activity in the mouse colon.

### 2.4. TRPC6 Regulates the Gut Microbiota

Decreased Zn^2+^ levels in the gastrointestinal lumen lead to a reduction in the diversity of gut microbiota and the proliferation of bacteria that thrive in low-Zn^2+^ environments, ultimately resulting in dysbiosis [12]. Additionally, it has been reported that intracellular Zn^2+^ plays a critical role in maintaining the morphology and function of Paneth cells, which are vital for gut microbiota maintenance [25,26,27]. Thus, Zn^2+^ is of utmost importance for the maintenance of gut microbiota in the gastrointestinal tract. The gut microbiota plays a major role in intestinal homeostasis, including redox states and the pathogenesis of colitis [28]. The expression of TRPC6 was observed in the intestinal mucosal layer and muscle layer (Figure 2E,F). To identify the specific bacteria that characterize the gut microbiota of TRPC6 KO mice, genomic DNA was extracted from feces collected at 12 weeks of age and subjected to 16S amplicon sequencing. Taxonomical analysis revealed no significant differences between WT and TRPC6 KO at the phylum level (Figure 4A). TRPC6 KO displayed an increase in *S24-7* at the family level (Figure 4B). At the genus level, *Anaerotrunus* was decreased in TRPC6 KO mice compared to WT mice (Figure 4C). At the species level, *Parabacteroides distasonisis* was enriched in TRPC6 KO mice (Figure 4D). These results suggest that TRPC6 regulates the intestinal gut microbiota.

### 2.5. Activation of TRPC6 Attenuates DSS-Induced Colitis

We finally examined whether the activation of TRPC6 improves colitis. Treatment with the TRPC6 activator (PPZ2 [29]) prevented DSS-induced weight loss, as well as disease activity index and colon shortening (Figure 5A–C). Additionally, treatment with PPZ2 significantly suppressed the expression of IL-6 in the colon (Figure 5D). Furthermore, PPZ2 treatment significantly increased Zn^2+^ amounts in the colon (Figure 5E). These results suggest that treatment with PPZ2 prevented the progression of DSS-induced colitis by maintaining Zn^2+^ homeostasis.

## 3. Discussion

Ion channels and transporters embedded in the cell membrane are essential for maintaining acid–base balance [30,31]. The gain and/or loss of ion channels and transporters leads to intestinal mucosal injury, such as bicarbonate and mucous layer destruction [32], epithelial cell loss [33], gut microbiota imbalance [34], and mucosal blood flow changes [35]. For example, the number of goblet cells and mucin-positive cells in the colon was reduced in sodium/hydrogen exchanger 8 KO mice, along with mucosal pH, MUC2 expression, as well as downregulated adenoma expression [32]. It is reported that the overexpression of the Piezo1 mechanosensitive channel or the stimulation of Piezo1 by Yoda1 decreased the tight junction protein claudin-1, leading to impaired colon epithelial integrity and the loss of barrier function [36]. Thus, ion channels and transporters play important roles that directly affect the mucosa, as well as tight junctions, microbial distribution, and mucosal blood flow. We observed an increase in TRPC6 during intestinal inflammation. It is known that TRPC6 expression is elevated in diabetic kidney disease, pulmonary hypertension, and cancer cells [37]. The TRPC6 upregulation in pulmonary arterial smooth muscle cells has been shown to be dependent on hypoxia-inducible transcription factor 1 (HIF-1) [38,39]. Intestinal inflammation in IBD is characterized by severe mucosal surface hypoxia and concomitant HIF stabilization [40,41]. The stabilization of HIF in IBD is attributed to alterations in the ratio of metabolic supply and consumption, resulting in inflammatory hypoxia. Consequently, TRPC6 expression may increase in a HIF1-dependent manner in intestinal inflammation. This study showed that some microbiota changed in TRPC6 KO mice. *Anaerotruncus* was decreased in TRPC6 KO mice. It has been reported that *Anaerotruncus* is a protective species against DSS-induced colitis in mice [42]. TRPC6 is reportedly expressed in intestinal gland cells, immune cells, smooth muscle cells, and neurons [43,44,45]. The gut microbiota is altered by mucus secreted from intestinal gland cells and IgA produced by immune cells [46,47]. TRPC6 has also been reported to be associated with intestinal motility [45]. Since the gut microbiota is also controlled by changes in peristalsis, TRPC6 expressed in smooth muscle may also contribute to altered expression patterns of the gut microbiota. Gut microbiota profiles reportedly change due to a deficiency of TRPA1 and TRPV1 channels, which are predominantly expressed in nerve cells [48]. It is also possible to consider that changes in gut microbiota profiles are due to neural control by TRPC6. Future studies should specify the cell type(s) that have the greatest impact on the microbial profile and colitis progression. TRP melastatin family TRP (TRPM) channels are reportedly activated via intestinal distension caused by bacterial infection [49]. In addition, gut microbiota metabolites modulate host physiology by activating some G protein-coupled receptors [50]. Since TRPC6 is activated by receptor stimulation and mechanical stimulation [17], identifying the gut bacteria or their metabolites that can increase TRPC6 channel activity may reveal new relationships between the gut microbiota and the host to protect against chronic inflammation.

We also found that TRPC6 regulates the expression of antioxidant proteins. Certain intestinal bacteria are known to induce antioxidant proteins via the Nrf2 pathway [51]. Furthermore, it has been reported that the constitutive activation of Nrf2 aggravates acute intestinal inflammation [24]. In other words, one possible reason for the worsening of DSS-induced colitis in TRPC6 KO mice might be the induction of reductive stress by antioxidant proteins. On the other hand, the mRNA expression levels of antioxidant proteins in the colon isolated from DSS-treated TRPC6 KO mice were found to be similar to those in the DSS-treated WT mouse colon. In DSS-induced colitis, the mucosal epithelium is damaged, leading to the infiltration of numerous inflammatory cells. Therefore, due to the altered composition of intestinal tissue cells in DSS-induced colitis compared to normal conditions, no difference was observed in the expression levels of antioxidant proteins between TRPC6 KO and WT mice. We found that TRPC3 is not directly associated with the development of colitis. TRPC3 and TRPC6 have been reported to be involved in cardiac oxidative stress. TRPC3 forms a complex with NADPH oxidase (Nox)2 and increases ROS [21,52]. On the other hand, the upregulation of the TRPC6 protein destabilizes the TRPC3-Nox2 complex and prevents hyperglycemia-induced ROS generation-dependent cardiac dysfunction [53]. Although TRPC3 is not permeable to Zn^2+^, TRPC6 is known to permeate various metal ions such as Zn^2+^ and Fe^2+^, in addition to Ca^2+^ and Na^+^ [20]. The selectivity of cations passing through TRPC channels might have profound effects on the intestine. Zn^2+^ is important for maintaining redox homeostasis and gut microbiota [11,12]. It has also been reported that Zn^2+^ deficiency increases oxidative stress and compensatory increases in antioxidant proteins [11,54]. Gut microbiota changes have been observed in Zn^2+^-deficient diets and Zip14, one of the Zn^2+^ transporters, KO mice [12,55]. Additionally, it is known that feeding Zn^2+^-deficient diets worsens colitis, while the administration of Zn^2+^ improves colitis [56,57]. The administration of a Zn^2+^ uptake inhibitor reduces *Anaerotruncus* in the gut microbiota [58]. Although TRPC6 alters not only Zn^2+^ homeostasis but also Fe^2+^ homeostasis [59], we could not observe any significant changes in lipid peroxidation or ferroptosis in TRPC6 KO mice. These observations suggest that among all the ions permeated through TRPC6, Zn^2+^ mainly contributes to intestinal homeostasis. However, little is known about the mechanism through which Zn^2+^ controls the gut microbial profile and redox status. Techniques for genetically modifying the gut microbiota [60,61] will help us to unravel the underlying molecular mechanism in the future. In IBD, Zn^2+^ levels in the body decrease. This decrease not only impairs the function of the gastrointestinal epithelium but also affects many immune cells. Furthermore, decreased Zn^2+^ concentration has been reported to increase inflammatory cytokines such as IL-1β and TNFα [62]. In Zn^2+^ deficiency, the proliferative response of T and B lymphocytes following IL-6 and IL-2 stimulation increases, while it adversely affects IL-4 signaling, leading to immune system impairment [63]. Since decreased Zn^2+^ levels are associated with worsening inflammation, Zn^2+^ supplementation may exert anti-inflammatory effects.

Anti-inflammatory drugs such as 5-aminosalicylic acid preparations, steroids, and immunosuppressants are used to treat IBD [4]. Recently, molecular target therapy using biological agents, including TNFα antibody, IL-12/23p40 antibody, and anti-α4β7 integrin antibody has also been performed [4,62]. Although it has become possible to control inflammation through drug treatment, IBD tends to recur easily and cannot be completely cured. After suppressing inflammation, it is considered necessary to continue to maintain the balance of the intestinal environment. Treatment methods aiming for a complete cure by transplanting stool from a healthy person are being attempted, but further research is believed to be necessary to optimize fecal microbiota transplantation [64]. Our results suggest that TRPC6-mediated Zn^2+^ influx contributes to normalizing gut microbial homeostasis in the IBD mouse model (Figure 6). Drugs that promote TRPC6-mediated Zn^2+^ influx may be a breakthrough strategy to prevent colitis progression by maintaining gut systemic homeostasis.

## 4. Materials and Methods

### 4.1. GEO Dataset Analysis

GEO RNA-sequencing Experiments Interactive Navigator [65] was used to retrieve normalized transcript levels from the GEO dataset (GSE83687 [66]). We analyzed the transcript levels of TRPC1-7 in intestinal tissue healthy control, CD patients, and UC patients.

### 4.2. Animals

All animal husbandry and experimental procedures were approved by the Ethics Committee of the Institute of Physiology or the Animal Welfare Committee of Kyushu University (protocol code: A23-027-1). Animal experiments were conducted following ARRIVE guidelines [67,68]; TRPC3 KO or TRPC6 KO mice (8–16 weeks old, male) were provided by Dr. Birnbaumer (NIEHS, USA) [21]. TRPC3 KO and TRPC6 KO mice were backcrossed onto 129/Sv mice background as described previously [20]. All mice were housed in individually ventilated cages with aspen woodchip bedding in groups of three mice per cage, under controlled environmental conditions (12 h light/12 h dark cycle, room temperature 21–23 °C, humidity 50–60%) with free access to standard food pellets and water.

### 4.3. DSS-Induced Colitis Model

Since 129/Sv mice are less susceptible to DSS-induced colitis, the administration of 5% (*w*/*v*) DSS to 129/Sv mice induces colitis in the mice [69,70]. WT, TRPC3 KO, or TRPC6 KO mice (10–15 weeks old) received 5% (*w*/*v*) dextran sulfate sodium (DSS) (mol wt, 36,000–50,000; ICN Biochemicals, Aurora, OH, USA) in drinking water for 6 days. The control group was given plain drinking water. Body weight, rectal bleeding, and stool hardness were measured daily as previously described [71]. Mice were euthanized by an overdose of isoflurane at the colon tissue harvest.

### 4.4. Disease Activity Index (DAI)

The DAI was used to assess the grade and extent of intestinal inflammation [71]. To quantify symptoms of colitis, body weight, stool hardness, and fecal occult blood were monitored daily using a previously published grading system [71]. Weight loss was scored as follows: 0, none; 1, 1–5%; 2, 5–10%; 3, 10–20%; 4, >20%. Diarrhea was scored as follows: 0, normal; 2, loose stools; 4, watery diarrhea. Blood in stool was scored as follows: 0, normal; 2, slight bleeding; 4, gross bleeding. The score of DAI ranges from 0 to 12 (total score).

### 4.5. PPZ2 Treatment

Male C57BL/6 mice (19–23 g, 8–10 weeks old) were obtained from Japan SLC, Inc. (Shizuoka, Japan). Male C57BL/6 mice were divided into two groups (n = 5 per group). All C57BL/6 mice were anesthetized with medetomidine (0.3 mg/kg; Kyoritsu Seiyaku Corporation, Tokyo, Japan), midazolam (4 mg/kg; Sandoz K.K., Tokyo, Japan), and butorphanol (5 mg/kg; Meiji Seika Pharma Co., Ltd., Tokyo, Japan) by i.p. injection; then, osmotic pumps (ALZET, Cupertino, CA, USA) for the sustained administration of PPZ2 (2.5 mg/kg/day) or vehicle were implanted intraperitoneally 5 days before DSS administration. After the operation, the surgical wound was sutured, and cefalexin was given to prevent infection. C57BL/6 mice are highly susceptible to DSS-induced colitis, and the administration of 2-3% (*w*/*v*) DSS to C57BL/6 mice causes colitis in mice [69,71,72]. For acute DSS-induced colitis, mice were given 3% (*w*/*v*) DSS in drinking water for 7 days. The mice were monitored daily for body weight, rectal bleeding, and stool consistency. Mice were euthanized by an overdose of isoflurane at the colon tissue harvest.

### 4.6. RNA Isolation and Quantitative Real-Time Reverse Transcription PCR

Distal colons were homogenized in a TRI reagent (Sigma-Aldrich, Saint Louis, MO, USA). The total RNA was extracted, and complementary DNA was synthesized as previously described [71]. Quantitative real-time PCR was performed as previously described [71]. The primers for mouse TRPC1 were forward primer 5′-GTCGCACCTGTTATTTTAGCTGC-3ʹ and reverse primer 5′-TGGGCAAAGACACATCCTGC-3′. The primers for mouse TRPC2 were forward primer 5′-CTCAAGGGTATGTTGAAGCAGT-3′ and reverse primer 5′-GTTGTTTGGGCTTACCACACT-3′. The primers for mouse TRPC3 were forward primer 5′-TCGAGAGGCCACACGACTA-3′ and reverse primer 5′-CTGGACAGCGACAAGTATGC-3′. The primers for mouse TRPC4 were forward primer 5′-CCAGAGCGAAGGTAATGGCAA-3′ and reverse primer 5′-GCTTAGGTTATGTCTCTCGGAGG-3′. The primers for mouse TRPC5 were forward primer 5′-GGGCTGAGACTGAGCTGTC-3′ and reverse primer 5′-TTGCGGATGGCGTAGAGTAAT-3′. The primers for mouse TRPC6 were forward primer 5′-AGCCAGGACTATTTGCTGATGG-3′ and reverse primer 5′-AACCTTCTTCCCTTCTCACGA-3′. The primers for mouse TRPC7 were forward primer 5′-CTTCCTGGACTCGGCTGAGTA-3′ and reverse primer 5′-GCGTTCTGCCCCATGTAGT-3′. The primers for mouse IL-6 were forward primer 5′-AAGGGCCAGGGATCTGTAAG-3′ and reverse primer 5′-TCTCTTGTTGCTCCCCAAAG-3′. The primers for mouse TNFα were forward primer 5′-ATGAGCACAGAAAGCATGATCCGC-3′ and reverse primer 5′-CCAAAGTAGACCTGCCCGGACTC-3′. The primers for mouse IL-1β were forward primer 5′-ATGGCAACTGTTCCTGAACTCAACT-3′ and reverse primer 5′-CAGGACAGGTATAGATTCTTTCCTTT-3′. The primers for mouse Nrf2 were forward primer 5′-CTGAACTCCTGGACGGGACTA-3′ and reverse primer 5′-CGGTGGGTCTCCGTAAATGG-3′. The primers for mouse SOD1 were forward primer 5′-AACCAGTTGTGTTGTCAGGAC-3′ and reverse primer 5′-CCACCATGTTTCTTAGAGTGAGG-3′. The primers for mouse Prdx3 were forward primer 5′-GGTTGCTCGTCATGCAAGTG-3′ and reverse primer 5′-CCACAGTATGTCTGTCAAACAGG-3′. The primers for mouse SRXN1 were forward primer 5′-CCCAGGGTGGCGACTACTA-3′ and reverse primer 5′-GTGGACCTCACGAGCTTGG-3′. The primers for mouse 18s ribosomal RNA (rRNA) were forward primer 5′-ATTAATCAAGAACGAAAGTCGCAGGT-3′ and reverse primer 5′-TTTAAGTTTCAGCTTTGCAACCATACT-3′. The cycling conditions were one cycle at 95 °C for 10 min, followed by 40 cycles of 30 s at 94 °C, 30 s at 60 °C, and 10 s at 72 °C. Additionally, 18s rRNA expression was used to normalize cDNA levels. The relative fold gene expressions of samples were calculated with the 2^−∆∆Ct^ method.

### 4.7. Zn^2+^ Concentration

Distal colons were homogenized in RIPA buffer containing 0.1% SDS, 0.5% sodium deoxycholate, 1% NP-40, 150-mM NaCl, 50-mM Tris–HCl (pH 7.4), and protease inhibitor cocktail (Nacalai Tesque, Kyoto, Japan). The Zn^2+^ concentration in the mouse distal colon was measured using the Metallo Assay kit Zinc LS (Metallogenics Co., Chiba, Japan), following the manufacturer’s protocol as described in a previous study [20]. Tissue lysate protein concentrations were determined using a bicinchoninic acid (BCA) assay kit (Nacalai Tesque, Kyoto, Japan). Subsequently, the Zn^2+^ concentration was normalized using the protein concentration, and the Zn^2+^ concentration of each sample was compared.

### 4.8. Immunohistochemistry

Immunofluorescent staining for frozen sections was performed as described previously [73]. Briefly, the distal colon was removed from the mice. For tissue sections, the distal colon was fixed in 4% paraformaldehyde. Frozen sections (5 μm thick) were cut and prepared for immunofluorescent staining. The expression of TRPC6 was detected using rabbit anti-TRPC6 antibody (1:200 dilution, Alomone Labs, Jerusalem, NY, USA, ACC-017, RRID:AB_2040243); TRPC6 immunoreactivity was detected using Alexa Fluor 488-conjugated goat anti-rabbit IgG antibody (1:1000 dilution, Thermo Fisher Scientific, Waltham, MA, USA, Cat# A32731, RRID:AB_2633280). Nonspecific immunoreactivity was blocked with 10% normal goat serum in PBS, 1% BSA, and 0.3% Triton X-100. After incubation with secondary antibodies, images were captured using a confocal laser scanning microscope (LSM 900, Zeiss, Oberkochen, Germany).

### 4.9. Gut Microbiota Analysis

The genomic DNA of the gut microbiota was extracted from mouse feces using the Nucleo-Spin^®^ DNA Stool (MACHEREY-NAGEL, Düren, Germany), as previously reported [74]. The DNA fragments of the variable V3-V4 region of the 16S rRNA gene were analyzed by Bioengineering Lab. Co., Ltd. (Kanagawa, Japan). Taxonomic analysis to detect differentially abundant taxa across groups was performed by uploading the taxonomic assignment table to the web-based tool MicrobiomeAnalyst [74].

### 4.10. Statistics

G*Power 3.1.9.2 software was used to calculate sample sizes for each group. All results are presented as mean ± SEM from at least five independent experiments and were considered significant if *p* < 0.05. Statistical comparisons were performed using the unpaired *t*-test for comparisons between two groups and using Tukey’s one-way analysis of variance with a post hoc test or Sidak’s two-way analysis of variance with a post hoc test for comparisons between three or more groups when F achievement values were *p* < 0.05, and there was no significant variation inhomogeneity. Statistical analyses were performed using GraphPad Prism 9.0 (GraphPad Software, LaJolla, CA, USA).

## Figures and Tables

**Figure 1 ijms-25-02401-f001:**
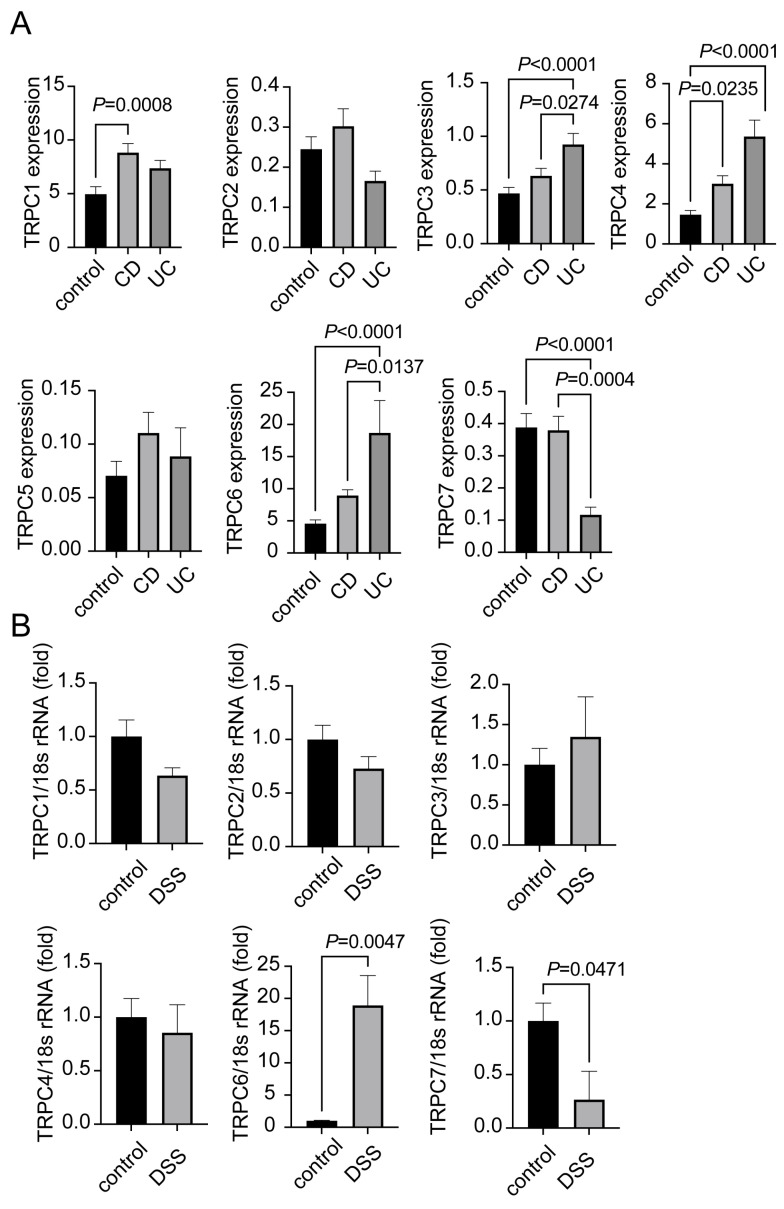
Increase in TRPC6 mRNA expression level in inflamed mucosa of IBD patients and colon tissue of DSS-treated mice: (**A**) The expression levels of TRPC1-7 genes were analyzed using a Gene Expression Omnibus (GEO) dataset (GSE83687) containing the expression profile of actively inflamed mucosa from colitis patients. Data are shown as the mean ± SEM (control; n = 60, CD; n = 42, UC; n = 32). *p* < 0.05, one-way ANOVA followed Tukey’s comparison test. (**B**) Quantification of TRPC1-7 mRNAs in colon tissue from each group of mice as measured by quantitative PCR and normalized against 18s rRNA (n = 5 mice in each group). Data are shown as the mean ± SEM; *p* < 0.05, significantly different as indicated; unpaired *t*-test.

**Figure 2 ijms-25-02401-f002:**
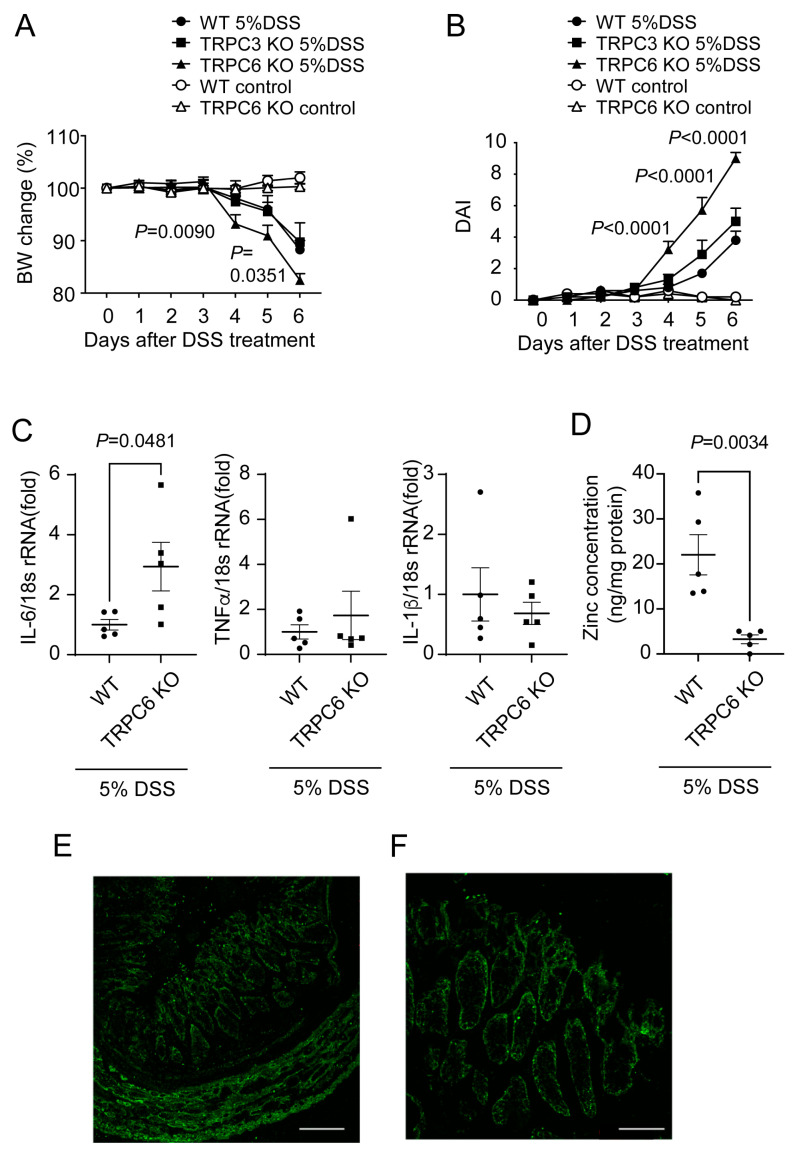
TRPC6 deficiency aggravates DSS-induced colitis progression: (**A**) Body weight changes and (**B**) disease activity index (DAI) in WT, TRPC3 KO, and TRPC6 KO mice treated with 5% DSS (n = 5 mice in each group). (**C**) Quantification of IL-6, TNFα, and IL-1β mRNAs in mouse colon tissues normalized by 18s rRNA (n = 5 mice in each group). (**D**) Zn^2+^ concentrations of colon tissues (n = 5 mice in each group). (**E**,**F**) Immunohistochemical staining of TRPC6 in the colon of mice. (**E**) Wide field of view. Scale bars; 100 μm. (**F**) Enlarged view. Scale bars; 50 μm. Data are shown as the mean ± SEM; *p* < 0.05, significance was determined using two-way ANOVA followed by Tukey’s comparison test (**A**,**B**) and unpaired *t*-test (**C**,**D**).

**Figure 3 ijms-25-02401-f003:**
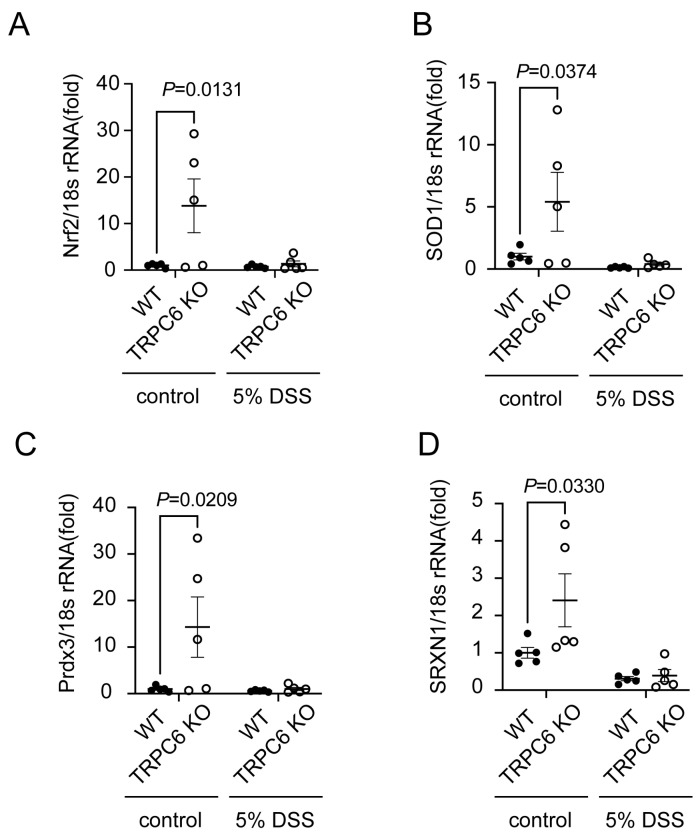
TRPC6 deficiency increased antioxidant proteins. Quantification of mRNA expression levels of antioxidants ((**A**) Nrf2; (**B**) SOD1; (**C**) Prdx3; (**D**) SRXN1) in mouse colon tissues (control WT: n = 5; control TRPC6 KO: n = 5; DSS WT: n = 5; DSS TRPC6 KO: n = 5). Data are shown as the mean ± SEM; *p* < 0.05, significance was determined using two-way ANOVA followed by Sidak’s comparison test.

**Figure 4 ijms-25-02401-f004:**
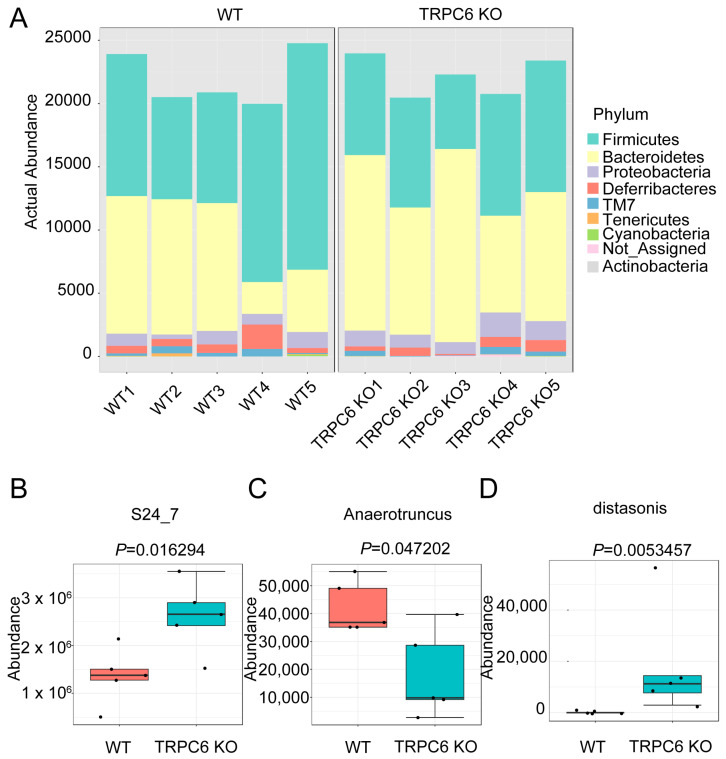
TRPC6 deficiency changes in the gut microbiota profile: (**A**) Taxonomic distribution at the phylum level, showing individual samples. Taxonomic changes in the intestinal microbiota. (**B**) Relative abundance of S24-7 is presented at the family level. (**C**) The relative abundance of Anaerotruncus is presented at the genus level. (**D**) The relative abundance of Parabacteroides distasonis is presented at the species level (n = 5 mice in each group). Taxa with LDA scores > 2.0 and *p* < 0.05, determined using the Wilcoxon signed-rank test, are shown.

**Figure 5 ijms-25-02401-f005:**
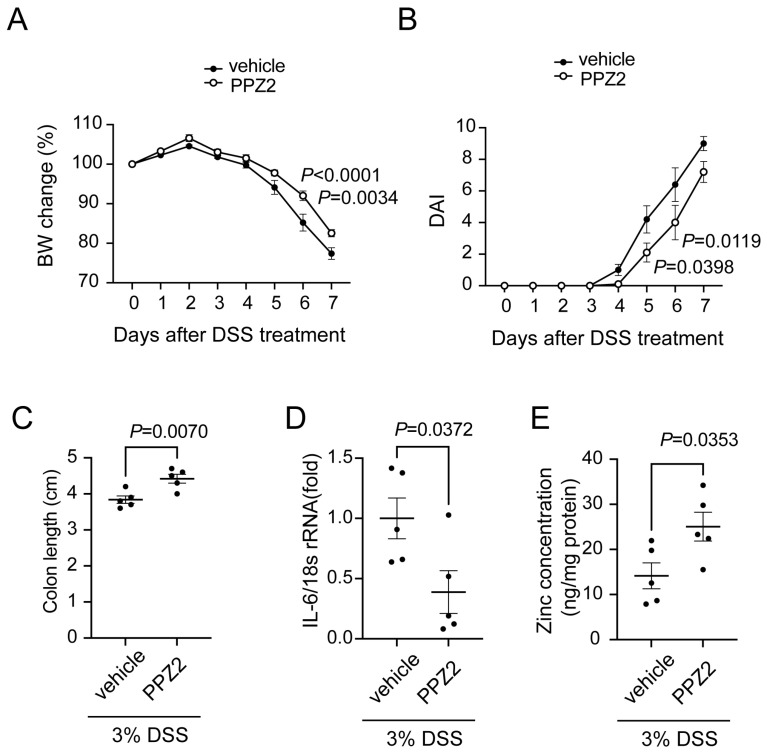
Treatment of mice with PPZ2 attenuates DSS-induced colitis. C57BL/6J mice were administered with DSS. Osmotic pumps including PPZ2 (2.5 mg/kg/day), or vehicle were implanted intraperitoneally 5 days before DSS administration: (**A**) Body weight changes, (**B**) DAI, and (**C**) colon length in mice treated with 3% DSS (n = 5 mice in each group). (**D**) Quantification of IL-6 mRNA in colon tissues (n = 5). (**E**) Zn^2+^ concentrations of colon tissues (n = 5 mice in each group). Data are shown as the mean ± SEM; *p* <0.05, significance was imparted using two-way ANOVA followed by Tukey’s comparison test (**A**,**B**) and unpaired *t*-test (**C**–**E**).

**Figure 6 ijms-25-02401-f006:**
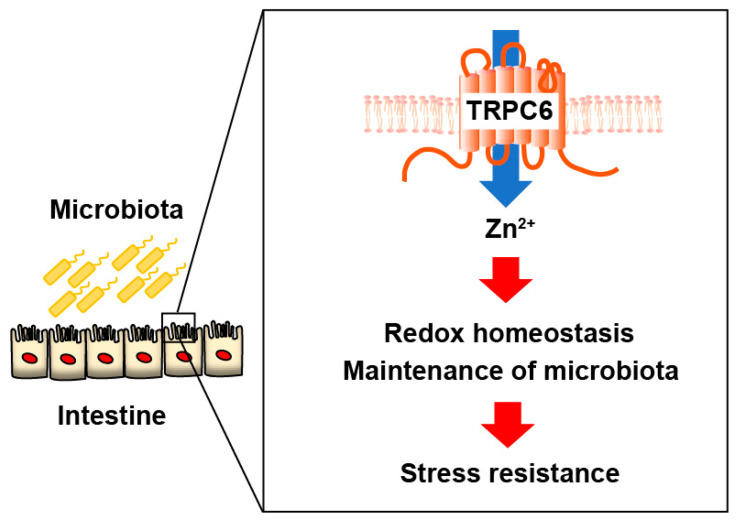
Schema of the role of TRPC6-mediated Zn^2+^ influx in the intestine. TRPC6-mediated Zn^2+^ influx plays a key role in stress resistance through the maintenance of redox homeostasis and gut microbiota.

## Data Availability

Data are contained within the article.

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
