# Peer review of "Pharmacological Activation of TRPC6 Channel Prevents Colitis Progression"

_ijms, 2024, doi:10.3390/ijms25042401_

Round 1
Reviewer 1 Report
Comments and Suggestions for Authors
1) Make corrections in the sentence. Zn2+ has been implicated in the regulation of intestinal redox and microbial homeostasis, 18 and we recently reported that transient receptor potential canonical (TRPC) 6 channel activity con- 19 tributes to intracellular Zn2+ homeostasis in mammalian cell.
2) Highlight the objective of the study.
3) Figure 1 can be more clearer.
4) Zn2+ is important for maintaining the gut microbiota. What is the mechanism.
5) Gain and/or loss of ion channels and transporters leads 183 to intestinal mucosal injury, such as bicarbonate and mucous layer destruction [28], epi- 184 thelial cell loss. Illustrate it.
6) Give more discussion to results.
7) Give citation to this, PPZ2 treatment
8) How zinc concentration was determined?
Comments on the Quality of English LanguageMinor editing is required
Author Response
Responses to the Editor and Reviewers
First of all, we truly appreciate your great efforts to review our manuscript thoughtfully. We have addressed the concerns raised by the reviewers by adding explanations to each of the queries. Our response to each point is presented below and we denote where modifications to the text in the manuscript have been made.
Responses to Reviewer 1.
1) Make corrections in the sentence. Zn2+ has been implicated in the regulation of intestinal redox and microbial homeostasis, 18 and we recently reported that transient receptor potential canonical (TRPC) 6 channel activity con- 19 tributes to intracellular Zn2+ homeostasis in mammalian cell.
[Response]
Thank you for your comment.
We changed words in the Abstract (yellow highlight).
2) Highlight the objective of the study.
[Response]
Thank you for your comment.
We described objective of the study in the Introduction (yellow highlight).
3) Figure 1 can be more clearer.
[Response]
Thank you for your comment. We modified the graph in Figure 1 to make it easier to see.
4) Zn2+ is important for maintaining the gut microbiota. What is the mechanism.
[Response]
Thank you for your comment. Decreased Zn2+ levels in the gastrointestinal lumen lead to a reduction in the diversity of gut microbiota and the proliferation of bacteria that thrive in low-zinc environments, ultimately resulting in dysbiosis. Additionally, it has been reported that intracellular Zn2+ plays a critical role in maintaining the morphology and the function of Paneth cells, which are vital for gut microbiota maintenance. Thus, Zn2+ is of utmost importance for the maintenance of gut microbiota in the gastrointestinal tract. We described these in the Results (yellow highlight).
5) Gain and/or loss of ion channels and transporters leads 183 to intestinal mucosal injury, such as bicarbonate and mucous layer destruction [28], epi- 184 thelial cell loss. Illustrate it.
[Response]
Thank you for your comment. For example, the number of goblet cells and mucin-positive cells in the colon was reduced in sodium/hydrogen exchanger 8 KO mice along with mucosal pH, MUC2 expression as well as downregulated in adenoma expression. It is reported that overexpression of Piezo1 mechanosensitive channel or stimulation of Piezo1 by Yoda1 decreased the tight junction protein claudin-1, leading to impaired colon epithelial integrity and loss of barrier function. We described these in the Discussion (yellow highlight).
6) Give more discussion to results.
[Response]
Thank you for your comment. We observed an increase in TRPC6 during intestinal inflammation. It is known that TRPC6 expression is elevated in diabetic kidney disease, pulmonary hypertension, and cancer cells. The TRPC6 upregulation in pulmonary arterial smooth muscle cells (PASMCs) has been shown to be dependent on hypoxia-inducible transcription factor 1 (HIF-1). Intestinal inflammation in IBD is characterized by severe mucosal surface hypoxia and concomitant HIF stabilization. The stabilization of HIF in IBD is attributed to alterations in the ratio of metabolic supply and consumption, resulting in inflammatory hypoxia. Consequently, TRPC6 expression may increase in a HIF1-dependent manner in intestinal inflammation. The mRNA expression levels of antioxidant proteins in the colon isolated from DSS-treated TRPC6 KO mice were found to be similar to those in DSS-treated WT mouse colon.In DSS-induced colitis, the mucosal epithelium is damaged, and there is infiltration of numerous inflammatory cells. Therefore, due to the altered composition of intestinal tissue cells in DSS-induced colitis compared to normal condition, it is thought that no difference was observed in the expression levels of antioxidant proteins between TRPC6 KO and WT mice. We described these in the Discussion (yellow highlight).
7) Give citation to this, PPZ2 treatment
[Response]
Thank you for your comment. We cited the paper in the Results (yellow highlight).
8) How zinc concentration was determined?
[Response]
Thank you for your comment. Distal colons were homogenized in RIPA buffer containing 0.1% SDS, 0.5% sodium deoxycholate, 1% NP-40, 150-mM NaCl, 50-mM Tris–HCl (pH 7.4), and protease inhibitor cocktail (Nacalai, Japan). The Zn2+ concentration in mouse distal colon was measured using the Metallo Assay kit Zinc LS (Metallogenics Co., Chiba, Japan) following the manufacturer’s protocol as described in a previous study [20]. Tisuue lysate protein concentrations were measured using a bicinchoninic acid (BCA) assay kit (Nacalai Tesque, Kyoto, Japan). The Zn2+ concentration was corrected by the protein concentration, and the Zn2+ concentration of each sample was compared. We described these in the Materials and Methods (yellow highlight).
Reviewer 2 Report
Comments and Suggestions for Authors
Nishida and colleagues studied the role of Zn2+ channel TRPC6 in the prevention of inflammatory bowel disease (IBD). They found that the levels of TRPC6 mRNA were increased in the bowels of IBD patients and of mice with artificially induced colitis. The knockout of TRPC6 in mice resulted in the increased severity of colitis symptoms and decreased Zn2+ content in colon. Moreover, TRPC6 KO abolished the induction of antioxidant genes expression in the colon in response to colitis induction. The treatment of mice with a specific TRPC6 activator prevented the development of artificially induced colitis and increased the levels of Zn2+ in colon. The authors conclude that the activity of TRPC6 prevents colitis progression. The presented data are both solid and novel. The authors should briefly discuss the available information about the potential mechanisms underlying the anti-inflammatory effects of zinc ions in the gut.
Author Response
Responses to the Editor and Reviewers
First of all, we truly appreciate your great efforts to review our manuscript thoughtfully. We have addressed the concerns raised by the reviewers by adding explanations to each of the queries. Our response to each point is presented below and we denote where modifications to the text in the manuscript have been made.
Responses to Reviewer 2.
Nishida and colleagues studied the role of Zn2+ channel TRPC6 in the prevention of inflammatory bowel disease (IBD). They found that the levels of TRPC6 mRNA were increased in the bowels of IBD patients and of mice with artificially induced colitis. The knockout of TRPC6 in mice resulted in the increased severity of colitis symptoms and decreased Zn2+ content in colon. Moreover, TRPC6 KO abolished the induction of antioxidant genes expression in the colon in response to colitis induction. The treatment of mice with a specific TRPC6 activator prevented the development of artificially induced colitis and increased the levels of Zn2+ in colon. The authors conclude that the activity of TRPC6 prevents colitis progression. The presented data are both solid and novel. The authors should briefly discuss the available information about the potential mechanisms underlying the anti-inflammatory effects of zinc ions in the gut.
[Response]
Thank you for your comment. In IBD, Zn2+ levels in the body decrease. This decrease not only impairs the function of the gastrointestinal epithelium, but also affects many immune cells. Furthermore, decreased zinc concentration has been reported to increase inflammatory cytokines such as IL-1β and TNFα. In Zn2+ deficiency, the proliferative response of T and B lymphocytes following IL-6 and IL-2 stimulation increases, while it adversely affects the IL-4 signaling, leading to immune system impairment. Since decreased Zn2+ levels are associated with worsening inflammation, zinc supplementation may exert anti-inflammatory effects. We described these in the Discussion (green highlight).